# Identification and Functional Analysis of a Defensin CcDef2 from *Coridius chinensis*

**DOI:** 10.3390/ijms23052789

**Published:** 2022-03-03

**Authors:** Tao Gong, Juan Du, Shang-Wei Li, Hai Huang, Xiao-Lang Qi

**Affiliations:** Guizhou Provincial Key Laboratory for Agricultural Pest Management of Mountainous Regions, Institute of Entomology, Guizhou University, Guiyang 550025, China; gongtao458@163.com (T.G.); juandudj@163.com (J.D.); hh118545@163.com (H.H.); 18286176698@163.com (X.-L.Q.)

**Keywords:** *Coridius chinensis*, antimicrobial peptide, defensin, innate immunity, antibacterial activity

## Abstract

*Coridius chinensis* belongs to Dinidoridae, Hemiptera. Previous studies have indicated that *C. chinensis* contains abundant polypeptides with antibacterial and anticancer activities. Antimicrobial peptides (AMPs), as endogenous peptides with immune function, play an indispensable role in the process of biological development and immunity. AMPs have become one of the most potential substitutes for antibiotics due to their small molecular weight and broad-spectrum antimicrobial activity. In this study, a defensin CcDef2 from *C. chinensis* was characterized based on bioinformatics and functional analyses. The mature peptide of CcDef2 is a typical cationic peptide composed of 43 amino acid residues with five cations, and contains three intramolecular disulfide bonds and a typical cysteine-stabilized αβ motif in defensins. Phylogenetic analysis showed that CcDef2 belongs to the insect defensin family. Analysis of gene expression patterns showed that *CcDef2* was expressed throughout developmental stages of *C. chinensis* with high levels at the nymphal stage and in adult tissues tested with the highest level in the fat body. In addition, the *CcDef2* expression was significantly upregulated in adults infected by bacteria. After expressed in *Escherichia coli* BL21(DE3) and renatured, the recombinant CcDef2 showed a significant antibacterial effect on three kinds of Gram-positive bacteria. These results indicate that CcDef2 is an excellent antibacterial peptide and a highly effective immune effector in the innate immunity of *C. chinensis*. This study provides a foundation for further understanding the function of *CcDef2* and developing new antimicrobial drugs.

## 1. Introduction

Multicellular organisms are constantly infected by pathogens and parasites in the surrounding environment, and the immune system can help them resist the invasion of bacterial pathogens. Insects have no specific immune system similar to vertebrate T-lymphocytes and B-lymphocytes, so innate immunity is the only way for insects to face pathogen infection [1,2]. Insect innate immunity mainly consists of humoral immunity and cellular immunity. Humoral immune responses include the production of antimicrobial peptides (AMPs), reactive intermediates of oxygen or nitrogen, and the complex enzymatic cascades that regulate coagulation or melanization of hemolymph [3,4,5,6]. Cellular responses refer to the immune response mediated by blood cells, including phagocytosis, nodulation, and encapsulation [7,8]. As a requisite part of the humoral immune mechanism, AMP is the first barrier of host defense and can kill bacteria, fungi, viruses, and protozoa or slow down their growth [9,10].

AMPs may be classified into four groups based on the differences in amino acid composition and structural characteristics: cecropin, defensin, glycine-rich peptides, and proline-rich peptides [11]. Cecropin was the first insect AMP that was isolated from the hemolymph of the pupae of *Hyalophora cecropia*. [12]. Since then, cecropins have been isolated from *Bombyx mori*, *Antherea pernyi*, Drosophila, and Sarcophagidae. The isolation of cecropin P1 from the small intestine of pigs indicated that cecropin may be widely present in animals [13]. The glycine-rich antimicrobial peptides were found in some insects, such as coleoptericin from *Allomyrina dichotoma* and hemiptericin from *Pyrrhocoris apterus* [14,15]. The proline-rich antibacterial peptides were reported in Italian bees [16,17] and Drosophila [18]. Defensin was found in polymorphonuclear neutrophils of mice and guinea pigs, initially named lysosomal cationic protein [19], and later defined and described as defensin in 1985 [20]. Two antibacterial peptides against Gram-positive bacteria isolated from larvae of *Phormia terranovae* were named insect defensins in 1989, based on their similarity to mammalian defensins [21,22]. Insect defensins are natural immune polypeptides produced by the fat body and hemolymph during accidental injury and invasion of pathogenic microorganisms, and are widely distributed in various insect orders such as Diptera, Hymenoptera, Odonata, Hemiptera, Lepidoptera, and Coleoptera. The defensins mainly act against Gram-positive bacteria, but some are also effective against Gram-negative bacteria or fungi. It has also been reported that insect defensins have antiviral and antitumor activities [23,24].

The emergence of antibiotic-resistant pathogens worldwide has become one of the severe threats to public health. The infection of drug-resistant bacteria is becoming more common and some pathogens are even resistant to various types of antibiotics. AMPs are considered as one of the most promising alternatives to antibiotics because of their particular antimicrobial mechanism by which bacteria are not easy to develop drug resistance [25]. As far as defensins are concerned, they can exert antibacterial effect by interacting with the negatively charged bacterial cell membrane, which has no specificity [26,27]. If bacteria are to be resistant to defensins, they must reshape the structure of their cell membrane. In addition, unlike the classical antibiotics that must penetrate the target cells to act on them, AMPs are thought to kill the target cells by destroying their cell membranes [28]. Theoretically, this mode of action will seriously reduce the resistance of microorganisms.

Our previous studies have revealed that the hemolymph of *C. chinensis* has broad-spectrum antibacterial activity that is caused mainly by AMPs in this insect [29]. In this study, a defensin *CcDef2* gene was identified from the full-length transcriptome of *C. chinensis* and verified by polymerase chain reaction (PCR) and Sanger sequencing. Sequence analysis and phylogenetic tree construction elucidated the evolutionary relationship between CcDef2 and other defensins. The spatiotemporal expression profile of *CcDef2* was analyzed at different developmental stages and in various tissues of adults by using real-time quantitative PCR (RT-qPCR). We analyzed the expression levels of *CcDef2* in adults injected with bacteria, which further revealed the role of CcDef2 in the innate immunity of *C. chinensis*. CcDef2 was successfully expressed in *E. coli* BL21 (DE3) and was shown to have the antibacterial effect by using a bacterial growth inhibition test. This study not only enriched the diversity of AMPs but also provided the theoretical feasibility for the development and utilization of AMPs in *C. chinensis.*

## 2. Results

### 2.1. Characteristic Analysis of CcDef2

The *CcDef2* cDNA is 859 bp in length (GeneBank accession numbers: MN816377), containing an ORF of 351 bp that encodes 116 amino acid residues (AAs), a 5′ untranslated region (UTR) of 38 bp, and a 3′ UTR of 470 bp UTR with a polyadenylation tailing signal (5′AATAAA 3′) (Figure 1 and Figure 2). CcDef2 contains a signal peptide of 17 AAs and two precursor peptides of 30 and 26 AAs at the N-terminus with three cleavage sites (A_17_↓I_18_, R_47_↓S_48_, and R_73_↓A_74_); therefore, its mature peptide consists of 43 AAs (Figure 2). The mature CcDef2 is a typical cationic peptide containing five positive charges, with a molecular weight of 4.70 kDa and a theoretical isoelectric point of 9.10. Structural analysis showed that the mature CcDef2 can form an α-helix and two β-pleated sheets as well as three intramolecular disulfide bonds (C_76_-C_107_, C_93_-C_112_, and C_97_-C_114_) formed by six cysteines. Homology modeling showed that the β-pleated sheet at the C-terminus of mature CcDef2 connected with the α-helix through two disulfide bonds to form a cysteine-stable αβ motif (CSαβ motif) and with the N-terminal via another disulfide bond to form a ring (Figure 3a). It is generally believed that the stable α-helix and β-sheet conformation is a functional structure for defensin to exert antibacterial activities. In addition, six cysteine residues and the eight other conserved AAs (Ala74, Thr75, Asp77, Ser80, Ala94, Gly104, Gly105, and Arg115) form a substrate-binding groove among the α-helix and β-sheets of mature CcDef2 (Figure 3b). CcDef2 also shows the distribution of positive charges (six basic amino acids) in the surface of the three-dimensional molecular structure (Figure 3c). Electrostatic surface analysis revealed that several regions in the surface of the solution structure were positively charged at a neutral pH (Figure 3c, in blue). Taken together, CcDef2 has the structural and electrostatic properties of insect defensins.

### 2.2. Multiple Sequence Alignment and Phylogenetic Analysis

Multiple sequence alignment indicated that insect and mammal defensins had six conserved cysteine residues, while plant defensins had eight ones. Interestingly, previous studies found that insect defensins were highly homologous with plant defensins [30]. Positions of cysteines between insect and plant defensins are similar but quite different from mammalian defensins (Figure 4). In addition, two typical characteristics of the mature peptides of most insect defensins are the presence of an alanine residue and a threonine residue (−AT−) at the N-terminus, and the presence of an arginine residue (−R−) at the C-terminus (Figure 4). CcDef2 and CcDef3 were highly homologous with 78.45% sequence similarity; however, CcDef2 shared only 46.55% similarity with CcDef and 44.83% with CcDef1 using ClustalW algorithm (CcDef and CcDef1, 94.12% similarity).

The phylogenetic tree shows that insect defensins group into a branch and different from mammal and plant defensins (Figure 5). Defensins from Diptera and Coleoptera cluster into a clade and those from Hemiptera and Hymenoptera fall into another clade. As shown in Figure 5, four defensins from *C. chinensis* form two subtypes. Interestingly, in a homology search it was found that CcDef2 and other homologues sequenced in insects also fell into two subtypes (https://www.ncbi.nlm.nih.gov, accessed 10 January 2022). During evolution, CcDef2 and CcDef3 have the closest homologue with defensin from *Pyrrhocoris apterus* (PaDef).

### 2.3. Spatiotemporal Expression Profile

The RT-qPCR results indicated that *CcDef2* was expressed throughout developmental stages of *C. chinensis*, with the highest level in the fifth-instar nymph, followed by the third- and fourth-instar nymphs, and with relatively low levels at other stages. There were no significant differences in *CcDef2* expression levels among the eggs, first- and second-instar nymphs, and female and male adults. The *CcDef2* expression level in the fifth-instar nymph was 180.70 times that in the egg and 2.66 times that in the third-instar nymph (Figure 6a). In general, AMPs are expressed in the fat body and secreted into the hemolymph to protect insects from the invasion of microorganisms. The *CcDef2* gene was expressed in the seven adult tissues tested, with higher expression levels in the fat body and hemolymph compared with the other tissues (Figure 6b). The highest expression of *CcDef2* in the fat body implies that it may be a very important immune tissue in innate immunity of *C. chinensis.*

### 2.4. Expression Patterns of CcDef2 upon Bacterial Challenge

The expression levels of *CcDef2* were detected by using RT-qPCR in adults challenged by *S. aureus* and *E. coli*. The *CcDef2* expression was upregulated 6–36 h after bacterial challenge and slightly declined at 48 h. The *CcDef2* expression level 12 h post injection was 56.89 times that without infection (Figure 7a). In the fat body from adults, we also observed a 40.28-fold increase in *CcDef2* expression level (Figure 7b). The changes implied that CcDef2 was involved in the immune response to bacterial infection.

### 2.5. Expression Analysis of Recombinant Protein

The nucleotide sequence of the mature CcDef2 was optimized according to the preference of codon usage in *E. coli*, and the optimized sequence is shown in Appendix A. By predicting, the theoretical molecular weight is about 18.2 kDa for the recombinant CcDef2 protein. SDS-PAGE displayed a target protein band at about 20 kDa in inclusion bodies recovered from *E. coli* cell lysates and this band was very close to the predicted molecular weight of the recombinant CcDef2 protein (Figure 8a), which was purified from the inclusion bodies by using the 6×His tag and Ni-NTA resin, a single clear band appearing at 20 kDa (Figure 8b). Western blot also showed a band of approximately 20 kDa (Figure 8c). These results indicated that the recombinant CcDef2 protein had been successfully expressed. The concentration of recombinant CcDef2 protein was 0.8 mg/mL based on the test result with the BCA method.

### 2.6. Antibacterial Spectrum and MIC

The results of antibacterial assay indicated that CcDef2 exhibited the antibacterial activities against the tested Gram-positive bacteria, i.e., *S. aureus*, *M. luteus*, and *B. subtilis*, but had no effects on the tested Gram-negative bacteria, i.e., *P. aeruginosa*, *E. coli*, and *S. typhi* (Figure 9). The findings are similar to those from most insect defensins. Remarkably, the recombinant CcDef2 had an obvious antibacterial effect on *S. aureus* compared with the controls. The minimum inhibitory concentration (MIC) of the recombinant CcDef2 against *S. aureus* was 50 μg/mL (0.92 μM) and the amount of bacteria incubated for 18 h was significantly lower than the control when the final concentration of the recombinant CcDef2 reached 6 μg/mL (Figure 10). MICs of the recombinant CcDef2 against *M. luteus* and *B. subtilis* were 1.24 μM and 1.56 μM, respectively (Table 1).

## 3. Discussion

Insect defensins were originally purified in *S. peregrina* [31]. They are antimicrobial peptides with potent activity against Gram-positive bacteria and weak activity against Gram-negative bacteria and fungi. So far, most defensins isolated and identified from insects are cationic peptides [32]. In this study, the cDNA sequence of *CcDef2* was cloned and confirmed by using RT-PCR and sequencing. The mature peptide of CcDef2 shares the similar sequence and structural characteristics as most insect defensins. Typical motifs of insect defensins are C-×5-16-C-×3-C-×9-11-C-×4-7-C-×1-C, which contain 34–51 amino acids [33]. Generally, six cysteines in an insect defensin form three intramolecular disulfide bonds that consist of Cys1-Cys4, Cys2-Cys5, and Cys3-Cys6. In addition, the β-pleated sheet of a mature insect defensin connects with the α-helix through two disulfide bonds to form a stable CSαβ motif. The CSαβ motif is an important spatial structure of defensins and is closely related to the structural stability and antibacterial activity [34,35,36]. The motif of CcDef2 is C-×16-C-×3-C-×9-C-×4-C-×1-C and is consistent with the typical one of a defensin. Significantly, CcDef2 is also a typical cationic peptide and its mature peptide contains five cations. Typical insect defensins are positively charged ones that can combine with negatively charged microbial cell membrane components through the mutual attraction to produce antibacterial effect [37].

The defensin gene found in *D. melanogaster* is single copy and lacks intron [38]. It was inferred that the insect defensins evolved independently. Subsequently, it was discovered that insect and mammalian defensins were different in disulfide bond connection and three-dimensional configuration, indicating that they were not homologous. Interestingly, Thevissen et al. demonstrated that defensins from insects and plants interacted with fungal glucosylceramides and they had high homology [39]. A variety of insect defensins indicate that insects have diverse immune mechanisms. At least two defensin homologues are usually found in many sequenced insect genomes or transcriptomes. In the transcriptome data of *C. chinensis*, we found four different defensin homologues that could be divided into two subtypes based on the phylogenetic analysis. CcDef2 and CcDef3 are highly homologous but different from CcDef and CcDef1, which may be the result of the finely tuned immune responses to counter pathogens and the adaptive evolution against pathogens in *C. chinensis*. Some studies suggested that the CSαβ motif of insect defensins evolved by gene duplication, followed by divergence due to selective evolutionary pressure, thus resulting in a diverse set of paralogues [40].

Previous studies have demonstrated that the expression pattern of insect AMP genes is regulated by their growth and development. The expression levels of three defensing genes (*AgDef2*, *AgDef3*, and *AgDef4*) from *A. gambiae* were almost undetectable in the eggs, pupae, and adults, but were high in the larvae [41]. Similarly, the *CcDef2* expression levels in the nymphs were much higher than those in the eggs and adults. The AMP *BhSGAMP-1* from *Bradysia hygida* was specifically expressed in the salivary glands of the larvae when they were preparing to molt, thus preventing microbial infection [42]. In this study, *CcDef2* was expressed in nymphs at high levels. We speculated that this may be due to the fact that an insect is more susceptible to infection by pathogens during molting, so *C. chinensis* nymphs improve the ability to resist the invasion of pathogens by increasing expression levels of *CcDef2*. In general, after infection or injury, insect AMPs are expressed in the fat bodies and secreted into hemolymph to protect insects from pathogenic microorganisms [43,44]. Similar to previous studies, the expression levels of *CcDef2* in the fat body were much higher than those in other tissues, indicating that the fat body is a vitally important immune tissue for *C. chinensis* in the process of resisting infection. Since the immune effectors are believed to be upregulated in response to infection, the mRNA expression levels of *CcDef2* were analyzed after bacterial challenge. AMP would appear in the hemolymph of infected insects about 6–12 h after infection [45]. As expected, the expression levels of *CcDef2* gene was significantly upregulated post bacterial challenge, suggesting that CcDef2 is closely associated with the immune response of *C. chinensis* against bacterial infection.

AMPs, known as the evolutionary ancient immune weapons, are the natural defense barriers of most organisms against the invasion of pathogens [46]. As a part of non-specific immune response, AMPs are directly involved in various immune responses to pathogens and perform broad-spectrum activity against various pathogenic microorganisms, such as Gram-positive bacteria, Gram-negative bacteria, enveloped viruses, and fungi [47]. Insect defensin, a main member of AMP, has always been an important subject in AMP research. At present, plenty of studies have been conducted to evaluate the antibacterial activity of insect defensins. The canonical antibacterial mechanism of defensins is that they interact with negatively charged bacterial membranes and insert into membrane bilayers to form pores, leading to membrane permeabilization and disruption [48]. In general, defensins are known to be active mainly against Gram-positive bacteria at various concentrations (MICs range from 0.4 μM to 100 μM) [49,50], but their activity against Gram-negative bacteria is weak because of the inability to penetrate the outer membrane of the bacteria [51,52,53,54,55]. Like most insect defensins, the recombinant CcDef2 produced inhibitory effects on Gram-positive bacteria (MICs range from 0.92 μM to 1.56 μM) but exhibited no inhibitory effects on such Gram-negative bacteria like *E. coli.* However, mammalian and plant defensins have been demonstrated to possess broad-spectrum antibacterial activity against both Gram-negative and Gram-positive bacteria [56,57,58,59]. AMPs diverge due to selective evolutionary pressure during the long-term evolution of insects and their functions also undergo corresponding fine-tuning and differentiation.

## 4. Materials and Methods

### 4.1. Insects and Bacteria

The laboratory populations of *C. chinensis* were collected from Guiyang, Guizhou, China, in 2019, and were reared at 28 ± 1 °C and a 75 ± 5% relative humidity under a 14:10 h light:dark photoperiod in the insectary of the Institute of Entomology, Guizhou University. Both nymphs and adults were fed with fresh pumpkin leaves. *Escherichia coli* BL21 (DE3) cells with 20% glycerol and expression vector pET-28b(+) (Merk KGaA, Damstadt, Germany) are kept at −80 °C in our laboratory. *E. coli* (ATCC 25922), *Salmonella typhi* (CMCC 50071), *Pseudomonas aeruginosa* (CMCC 10104), *Micrococcus luteus* (CMCC 28001), *Bacillus subtilis* (CMCC 63501), and *Staphylococcus aureus* (ATCC 25923) are kept in our laboratory.

### 4.2. RNA Extraction and Gene Cloning

An HP Total RNA Kit (Omega Bio-Tek Inc., Norcross, GA, USA) was used to extract the total RNA from different developmental stages (the eggs, first–fifth-instar nymphs, and females and males) and from various tissues (the head, integument, fat body, hemolymph, testis, ovary, and muscle) of *C. chinensis* adults. The hemolymph was extracted by using the double-tube centrifugation [60]. The purity and concentration of RNA were determined with a NanoDrop 2000 spectrophotometer (Thermo Fisher Scientific, Waltham, MA, USA) and the quality of RNA was detected by using 1% agarose gel electrophoresis. RNA was used as a template to synthesize cDNA using a RevertAid First Strand cDNA Synthesis Kit (Thermo Fisher Scientific, Waltham, MA, USA). The synthesized cDNA was diluted to 400 ng/μL and stored at −20 °C. Based on the transcriptome sequence information, primers for the reverse transcription-PCR (RT-PCR) primers were designed with Primer Premier 6.0 (PREMIER Biosoft, Palo Alto, CA, USA) (Table 2). RT-PCR was conducted in a 50-μL reaction system containing 2 μL of cDNA, 1 μL each of forward and reverse primers, 25 μL of 2× Taq PCR StarMix (GenStar, Beijing, China), and 21 μL of sterile deionized water in a T100 Thermal Cycler (Bio-Rad, Hercules, CA, USA). The PCR reaction parameters were as follows: 95 °C for 2 min; 30 cycles of 95 °C for 30 s, 56 °C for 30 s, and 72 °C for 1 min; and a final extension of 72 °C for 5 min. PCR products were detected by using 1% agarose gel electrophoresis, recovered by a SanPrep Column DNA Gel Extraction Kit (Sangon Biotech, Shanghai, China), and then ligated into the pMD18-T vector. The ligation products were transformed into *E. coli* Top10 competent cells for sequencing.

### 4.3. Bioinformatic Analyses

The open reading frame (ORF) of *CcDef2* was predicted with ORFfinder (https://www.ncbi.nlm.nih.gov/orffinder) (accessed on 6 April 2021) and homology was searched by BLAST against the NCBI “nr” protein database (https://blast.ncbi.nlm.nih.gov/Blast.cgi) (accessed on 6 April 2021). The molecular weight and isoelectric point of the CcDef2 peptide were predicted with the ExPaSy ProtParam tool (https://web.expasy.org/protparam) (accessed on 6 April 2021). Dianna 1.1 server (http://clavius.bc.edu/clotelab/DiANNA) (accessed on 6 April 2021) was used to predict disulfide bonds. The phylogenetic tree was constructed by using the neighbor-joining method in the MEGA X software with 1000 replicates. Three-dimensional structure of the mature CcDef2 was predicted with SWISS-MODEL (https://swissmodel.expasy.org) (accessed on 15 March 2021) based on homologous modeling and then its molecular graph was drawn using PyMOL 2.5 (Schrodinger, New York, NY, USA).

### 4.4. Spatiotemporal Expression Profile

The mRNA expression levels of *CcDef2* at different developmental stages and in various adult tissues were detected by using the CFX96 Real-Time PCR Detection System (Bio-Rad, Hercules, CA, USA). Specific primers specific to *CcDef2* for RT-qPCR were designed by Primer Premier 6.0 (PREMIER Biosoft, Palo Alto, CA, USA). RT-qPCR was carried out in a 20-μL reaction system that contained 1 μL of cDNA, 1 μL each of forward and reverse primers, 10 μL of 2× SYBR Select Master Mix (Thermo Fisher Scientific, Waltham, MA, USA), and 7 μL of sterile deionized water. RT-qPCR was performed with the following parameters: 50 °C for 2 min; 95 °C for 2 min; and 40 cycles of 95 °C for 15 s, 60 °C for 1 min. PCR products were verified by dissociation curve analysis. The *β-actin* gene of *C. chinensis* (GenBank accession number: MK370101) was used as the internal control, and the primers used are listed in Table 2. These experiments were repeated three times for each sample.

### 4.5. Bacterial Infection Assay

*S. aureus* and *E. coli* were inoculated into 20 mL Luria-Bertani (LB) medium and cultured in an incubator at 200 rpm at 37 °C. When the OD_600_ value of the culture reached 0.1, the bacteria were centrifuged at 10,000× *g* for 5 min, and collected and washed with phosphate-buffered saline (PBS, pH 7.5). These two bacteria were mixed and then suspended in PBS to make OD_600_ = 0.01. One hundred healthy *C. chinensis* adults were randomly selected for intraperitoneal injection using 1 μL of the bacterial suspension. These adults were gathered at 6, 8, 12, 24, 36, 48, 60, and 72 h after infection and then frozen with liquid nitrogen for standby. The fat bodies were dissected from the adults at 12 h postinfection. Healthy adult samples were infected with PBS as the control group. Three repeated experiments were performed. Total RNA extraction, cDNA synthesis, and RT-qPCR were performed as described above.

### 4.6. Construction of Recombinant Expression Vector

Due to the small molecular weight of the CcDef2 mature peptide, we employed two flexible linkers (−GGGGSGGGGSGGGGS−) to connect three mature CcDef2 for heterologous expression in order to facilitate the purification and identification of the expressed products. The GenSmar Codon Optimization Tool (Version Beta 1.0) (https://www.genscript.com/tools/gensmart-codon-optimization) (accessed on 5 May 2021) was used to optimize the nucleotide sequence of the mature CcDef2 according to the preference of codon usage in *E. coli*. The optimized sequence was synthesized (Sangon Biotech, Shanghai, China) and then cloned into the pET-28b(+) vector. The recombinant plasmid, pET-28b(+)-CcDef2, was transformed into *E. coli* TOP10. The bacteria were spread onto LB agar plates containing 50 μg/mL kanamycin and incubated at 37 °C overnight. Positive clones were identified by three steps, namely PCR, digestion with *Nde* I and *Xho* I, and sequencing.

### 4.7. Heterologous Expression, Purification and Refolding

The successfully constructed recombinant plasmid was transformed into *E. coli* BL21 (DE3) cells. The bacteria were cultured overnight at 37 °C and a single colony was inoculated into 50 mL LB liquid medium. When the OD_600_ value of the bacterial solution reached 0.6, a final concentration of 0.5 mM IPTG was added. Then the strains were cultured at 20 °C in a concussive manner for 24 h and harvested. The bacteria were precipitated by centrifugation at 4 °C and 4000× *g* for 10 min. The bacterial precipitate was resuspended in PBS (pH 7.5) and fragmented by ultrasound on ice. After 20 min of centrifugation at 4 °C and 10,000× *g*, the cell disruption precipitate was collected and dissolved in denaturation buffer (pH 7.5 PBS and 8 M urea). The protein samples were detected by sodium dodecyl sulfate-polyacrylamide gel electrophoresis (SDS-PAGE) and then visualized by staining with Coomassie brilliant blue.

According to the above conditions, the expression strains were inoculated into 1000 mL LB medium and massively cultured. The expressed protein with a 6×His-tag at the N-terminus was purified using a His-tag Protein Purification Kit (Denaturant-resistant) (Beyotime Biotechnology, Shanghai, China) following the manufacturer’s instructions. The concentration of the purified CcDef2 was measured with a BCA Protein Assay Kit (Solarbio, Beijing, China) and 20 μL of CcDef2 was subjected to SDS-PAGE. The recombinant protein was identified by Western blot analysis using rabbit anti-His polyclonal antibody (1:800 dilution) as the primary antibody and horseradish peroxidase-conjugated goat anti-rabbit IgG (1:5000 dilution) as the secondary antibody. The purified CcDef2 protein was dialyzed for 12 h at 4 °C in a dialysis bag against 0.4 mM oxidized glutathione, 4 mM reduced glutathione, and decreasing concentrations of urea (4, 2, 1, and 0.1 M). The refolded recombinant CcDef2 was concentrated and stored in Tris-NaCl buffer (50 mM Tris and 300 mM NaCl, pH 8.0).

### 4.8. Antimicrobial Assay

The antibacterial activity of CcDef2 was determined using the agar plate diffusion method. The inclusion body protein was used as a blank control. Six kinds of bacteria including *E. coli*, *S. typhi*, *P. aeruginosa*, *M. luteus*, *B. subtilis*, and *S. aureus* were picked out with sterilized toothpicks and inoculated into sterile centrifuge tubes containing 1 mL sterile LB medium without antibiotics, respectively. The bacteria were cultured for 12 h at 200 rpm and 37 °C. The 100 mL MH agar medium was sterilized for 30 min at 121 °C and then cooled to 50 °C. Ten microliters of bacterial solution was mixed with 20 mL of MH agar medium (1:2000) and poured into a 90 mm petri dish. Holes with a diameter of 6 mm were drilled in the agar plate using a puncher, and then CcDef2 (30 μg) and the inclusion body protein (30 μg) were separately added into these holes. After standing at room temperature for 2 h, the plates were incubated at 37 °C for 24 h and photographed using a ChemiDoc MP Imaging System (Bio-Rad, Hercules, CA, USA). All tests were independently repeated for three times.

### 4.9. MIC Determination of CcDef2

*S. aureus* was inoculated onto an MH agar plate for activation. Single colony that grew well on the plate was selected and inoculated into the MH broth. The bacteria were cultured at 37 °C and 200 rpm to the logarithmic growth phase (about 10 h) and then harvested by centrifugation at 5000 rpm for 5 min. The strains were washed with sterilized MH broth and diluted to 10^5^ CFU/mL. The MIC of CcDef2 was determined using the microdilution method. The refolded recombinant CcDef2 (0.8 mg/mL) was diluted to different concentrations with Tris-NaCl buffer at pH 8.0. Aliquots (100 μL) of the serial dilutions were dispensed into a 96 well microtiter plate and mixed with 100 μL of the bacterial cultures, and Tris-NaCl buffer instead of protein solution was set as the control. The microtiter plate was incubated at 37 °C for 18 h and bacterial growth was measured as absorbance at 600 nm using a Multiskan GO plate reader (Thermo Fisher Scientific, Waltham, MA, USA). MICs of CcDef2 to the other five kinds of bacteria (*E. coli*, *S. typhi*, *P. aeruginosa*, *M. luteus*, and *B. subtilis*) were detected using the same method as above. All tests were independently repeated for three times.

### 4.10. Data Analysis

The expression levels of *CcDef2* were calculated using the 2^−ΔΔCt^ method at different developmental stages, in various adult tissues, and in adults induced by bacteria. Data were analyzed using SPSS 22.0 statistical software (SPSS Inc., Chicago, IL, USA), and multiple comparisons were performed using one-way analysis of variance (ANOVA) and Duncan’s multiple range test. A *p*-value less than 0.05 was considered as statistical significance.

## 5. Conclusions

In this study, *CcDef2* from *C. chinensis* was characterized by using molecular cloning procedures, heterologous expression, and functional analysis. We found that the mature CcDef2 has the similar sequence and structural characteristics as most insect defensins, indicating that CcDef2 is conserved during evolution. CcDef2 possesses an excellent inhibitory effect on Gram-positive bacteria and is a highly effective immune effector in the innate immunity of *C. chinensis*. This study provides a theoretical feasibility for the development of AMPs in *C. chinensis*.

## Figures and Tables

**Figure 1 ijms-23-02789-f001:**
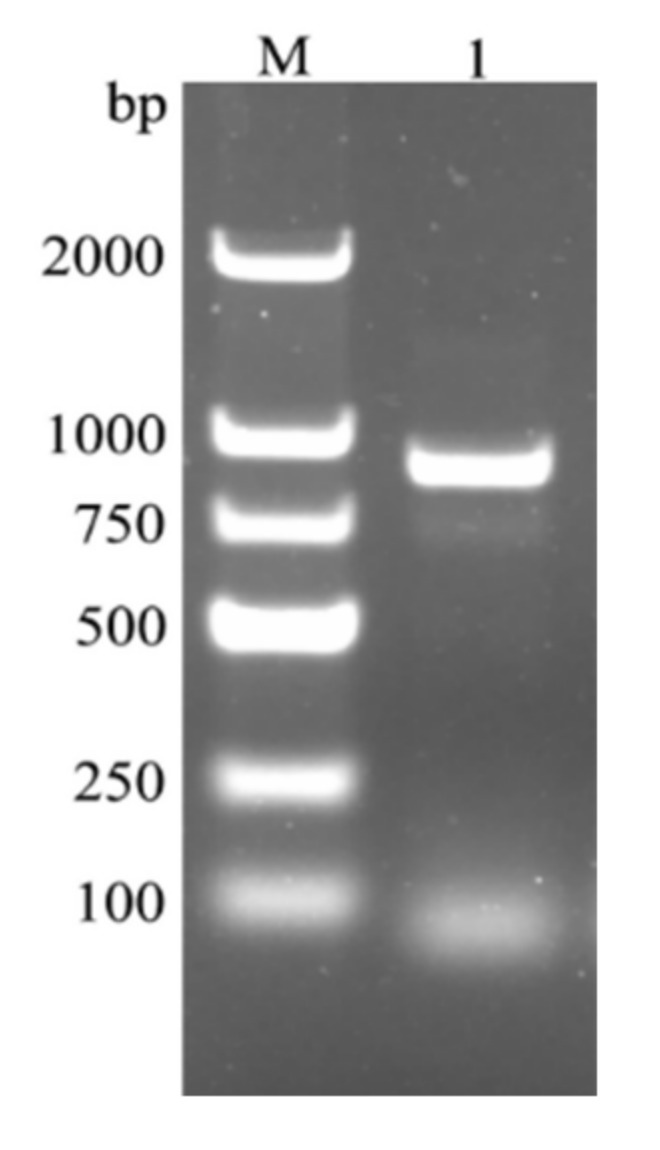
PCR amplification product of *CcDef2* from *C. chinensis*. M: DL2000 DNA marker. 1: PCR product of *CcDef2*.

**Figure 2 ijms-23-02789-f002:**
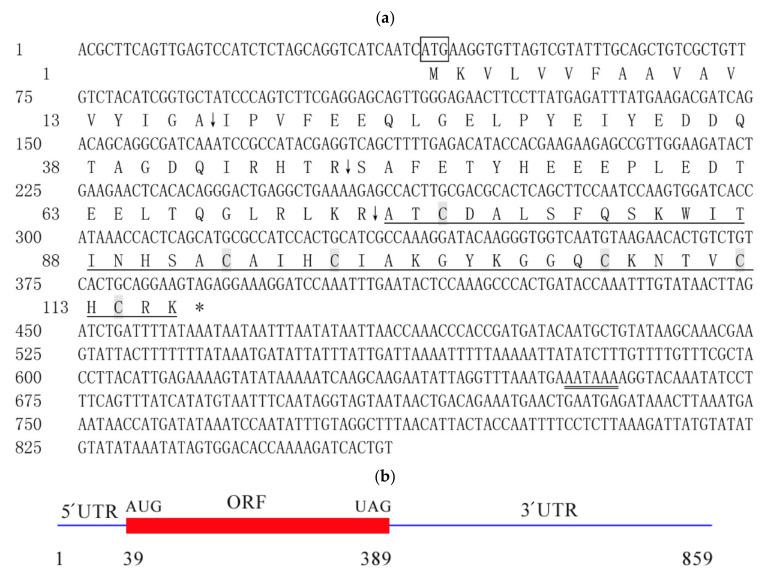
Nucleotide sequence, mRNA, and deduced amino acid sequence of *CcDef2*. (**a**) Nucleotide and amino acid sequences of *CcDef2*. (**b**) Schematic diagram of the *CcDef2* mRNA. The amino acid sequence of the mature CcDef2 is marked by a single underline. The star codon (ATG) is marked by a box. The polyadenylation signal is indicated with a double underline. The asterisk indicates the stop codon (TAG). Vertical arrows indicate the cleavage sites of signal peptide and precursor peptide. Six cysteines are marked with gray shade. The 5′ untranslated region (UTR) contains the 1st to 38th nucleotides and the 3′ UTR contains the 390th to 859th nucleotides.

**Figure 3 ijms-23-02789-f003:**
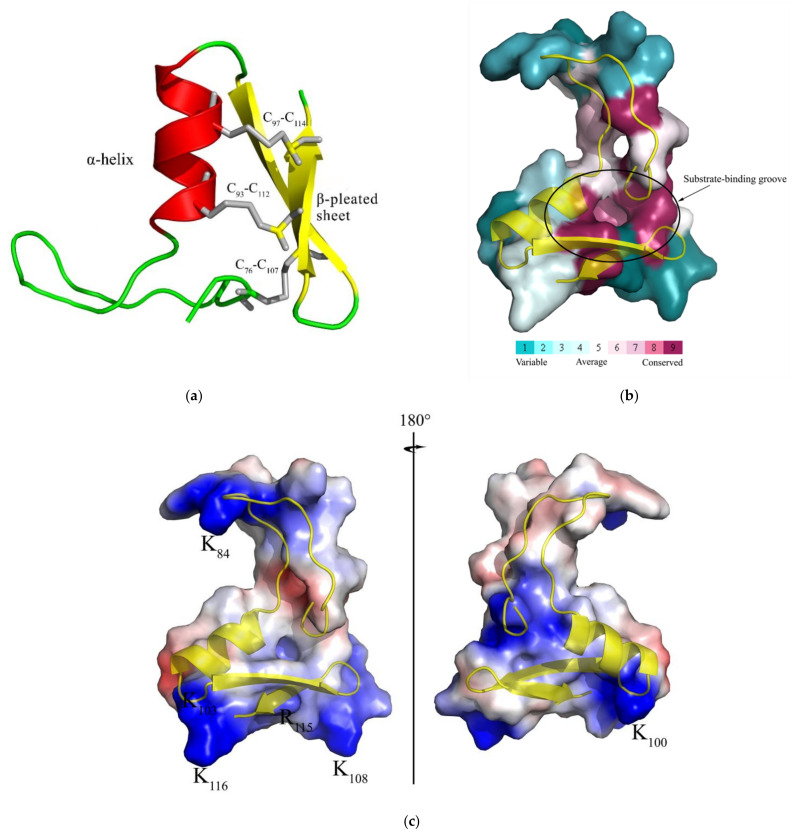
Three-dimensional molecular structure of the mature CcDef2. (**a**) This figure is generated with PyMOL 2.4 based on the CcDef2.pdb data. C_76_-C_107_, C_93_-C_112_, and C_97_-C_114_ indicate disulfide bonds. (**b**) The evolutionary conservation of amino acid positions in the mature CcDef2 based on the phylogenetic relations between homologous sequences. Conserved residues are displayed in fuchsin in the structure. Homology modeling was performed with the ConSurf software (https://consurf.tau.ac.il/) (accessed on 15 March 2021) and optimized by using PyMOL 2.4. (**c**) Electrostatic potential map of the mature CcDef2. The positively charged regions and negatively charged regions are shown in blue and red, respectively.

**Figure 4 ijms-23-02789-f004:**
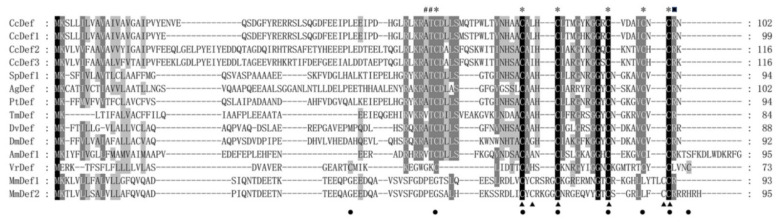
Multiple sequence alignment of four defensins from *C. chinensis* and typical defensins from other species. These species include seven insects: *Sarcophaga peregrina* (SpDef1), P18313; *Anopheles gambiae* (AgDef), Q17017; *Protophormia terraenovae* (PtDef), P10891; *Tenebrio molitor* (TmDef), Q27023; *Drosophila virilis* (DvDef), AHW49172; *Drosophila melanogaster* (DmDef), P36192 and *Apis mellifera* (AmDef1), P17722; a plant *Vigna radiata* (VrDef), AAR08912 and two mammals *Mus musculus* (MmDef1, NP_034161; and MmDef2, NP_001182563). Asterisks indicate positions of cysteines in insect defensins, solid triangles represent positions of cysteines in mammalian defensins, and solid circles denote positions of cysteines in plant defensins. #: Pounds indicate alanine (A) and threonine (T), and the black box marks arginine (R).

**Figure 5 ijms-23-02789-f005:**
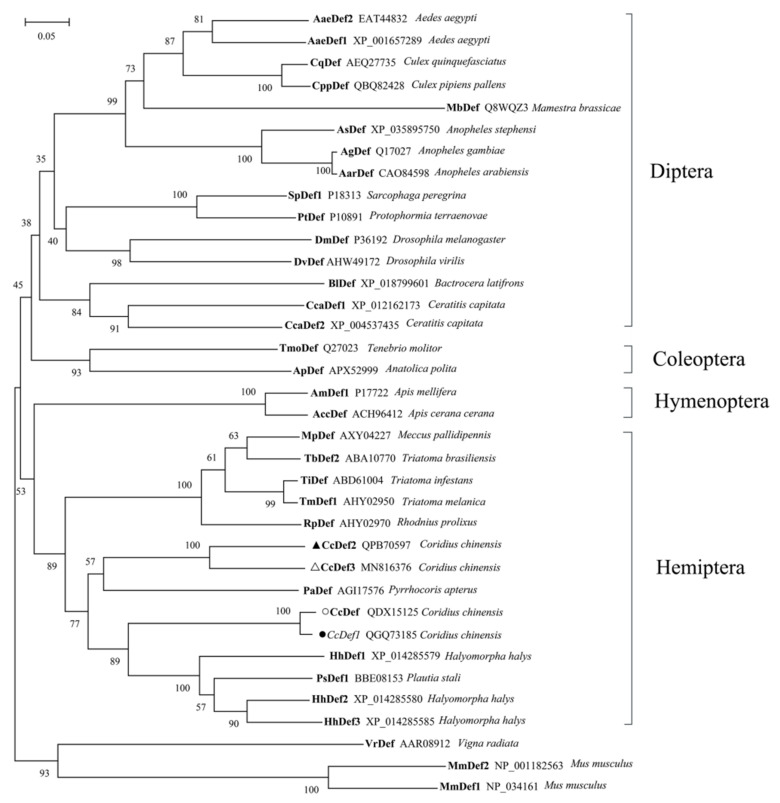
A cluster dendrogram of 36 defensins from 28 species. This tree was constructed using MEGA X based on the neighbor-joining method (NJ). One thousand replicates were performed and bootstrap confidence values are shown at the nodes of this tree. Defensins from *Vigna radiata* (VrDef) and *Mus musculus* (MmDef1 and MmDef2) are used as outgroups. Four defensins from *C. chinensis* are marked with a filled circle, a circle, a filled triangle, and a triangle, respectively.

**Figure 6 ijms-23-02789-f006:**
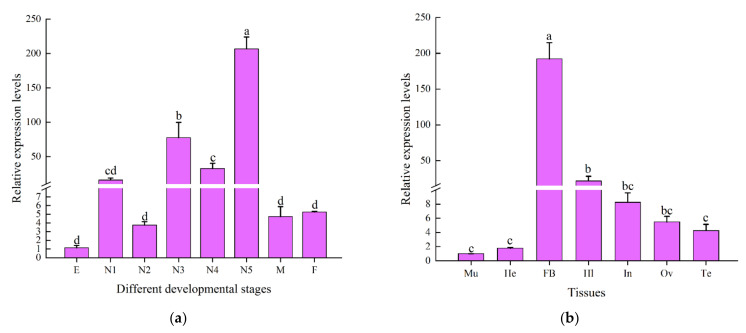
Relative expression levels of *CcDef2* at different developmental stages (**a**) and in various tissues of *C. chinensis* adults (**b**). (**a**) E: egg; N1–N5: first–fifth-instar nymphs; M: male; F: female. The adults come from different male and female individuals. (**b**) Mu: muscle; He: head; FB: fat body; Hl: hemolymph; In: integument; Ov: ovary; Te: testis. Values are the mean ± SD of three replicates. Differences between groups were analyzed using one-way ANOVA and Duncan’s multiple range test. Different lowercase letters above the columns indicate significant differences (*p* < 0.05, Duncan’s test).

**Figure 7 ijms-23-02789-f007:**
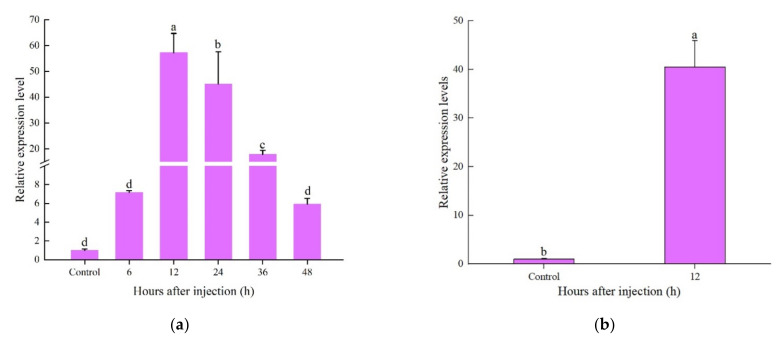
Expression levels of *CcDef2* at different times upon bacterial infection. (**a**) *C. chinensis* adults. (**b**) The fat body from adults. Healthy adults were infected with phosphate-buffered saline (PBS) as the control. Data are represented as the mean ± SD of three replicates. Differences between groups were analyzed using one-way ANOVA and Duncan’s multiple range test. Different lowercase letters above the columns indicate significant differences (*p* < 0.05, Duncan’s test).

**Figure 8 ijms-23-02789-f008:**
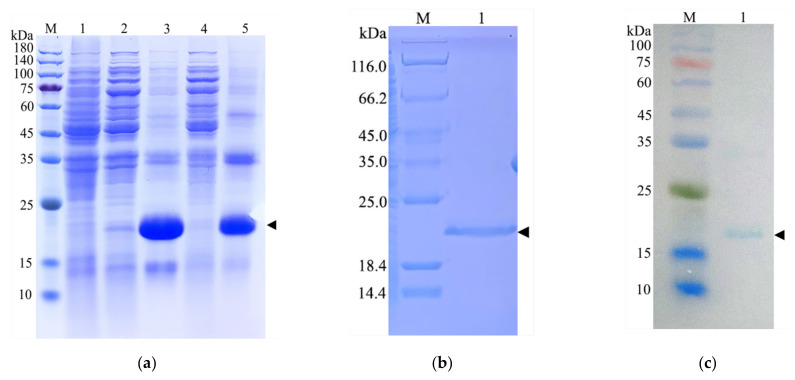
Identification of the recombinant CcDef2 protein. (**a**) SDS-PAGE analysis of the recombinant CcDef2 expressed by *E. coli* induced. Lane 1: the non-induced bacterial culture. Lanes 2 and 4: the soluble supernatant. Lines 3 and 5: inclusion bodies. (**b**) Purification of the recombinant CcDef2. Lane 1: fusion CcDef2. (**c**) Verification of the recombinant CcDef2 by Western blot. Lane 1: the blotting band of the fusion CcDef2 protein. M: Protein molecular weight markers. The arrowheads indicate the bands of the recombinant CcDef2 proteins.

**Figure 9 ijms-23-02789-f009:**
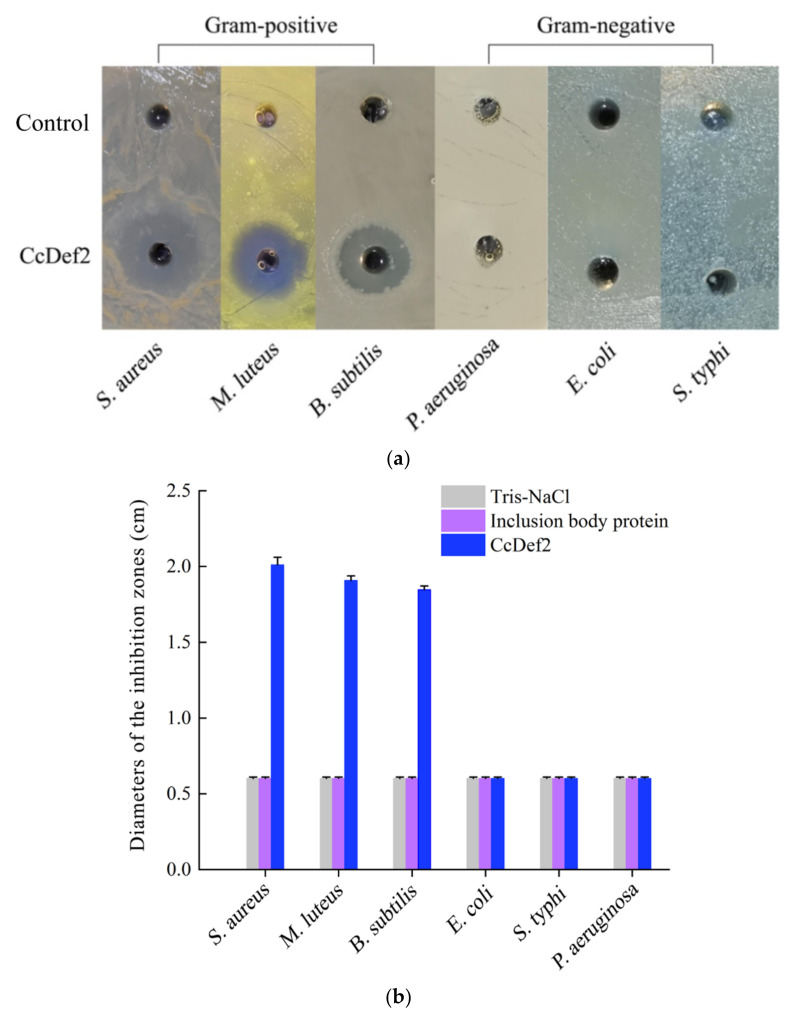
Antibacterial activity of the recombinant CcDef2. (**a**) Inhibition zones of CcDef2 against six kinds of bacteria. (**b**) Diameters of the inhibition zones. Tris-NaCl buffer was used as the control. The non-refolded inclusion body was used as a negative control. For the plate without a lytic zone, the diameter of a hole was used to represent the one of a lytic zone (0.6 ± 0.01 cm). Data are expressed as the mean ± SD of three replicates.

**Figure 10 ijms-23-02789-f010:**
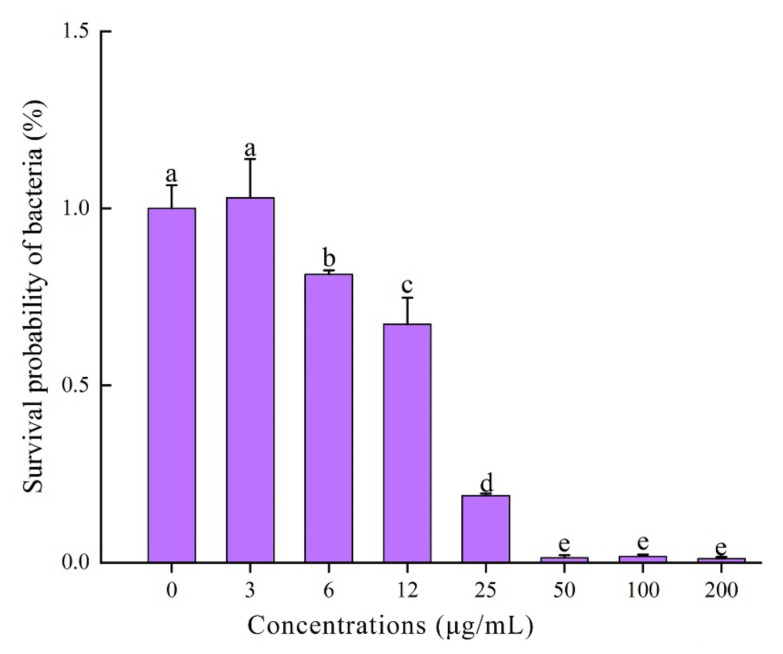
The MIC of the recombinant CcDef2 against *S. aureus*. Data are expressed as the mean ± SD of three replicates. Differences between groups were analyzed using one-way ANOVA and Duncan’s multiple range test. Different lowercase letters above the columns indicate significant differences (*p* < 0.05, Duncan’s test).

**Table 1 ijms-23-02789-t001:** Minimum inhibitory concentrations of the recombinant CcDef2.

Bacterial Strain	MIC (μM)
Gram-positive bacteria	
*S. aureus* ATCC 25923	0.92
*M. luteus* CMCC 28001	1.24
*B. subtilis* CMCC 63501	1.56
Gram-negative bacteria	
*E. coli* ATCC 25922	ND
*P. aeruginosa* CMCC 10104	ND
*S. typhi* CMCC 50071	ND

Notes: ND, no detectable activity. Data are expressed as the mean.

**Table 2 ijms-23-02789-t002:** Primers for verification and expression analyses of *CcDef2*.

Primer Name	Sequence	Primer Usage
Def2-F	5′-ACGCTTCAGTTGAGTCCATCT-3′	Cloning
Def2-R	5′-ACAGTGATCTTTTGGTGTCCACT-3′	
Def2-qF	5′-GCTGTCGCTGTTGTCTACATCGGT-3′	RT-qPCR
Def2-qR	5′-CGGCTCTTCTTCGTGGTATGTCTC-3′	
Actin-F	5′-ACCGCTGAGAGGGAAATCG-3′	
Actin-R	5′-CAAGAAGGAAGGCTGGAAGAG-3′	

## Data Availability

Not applicable.

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
