# Peer review of "Identification and Functional Analysis of a Defensin CcDef2 from *Coridius chinensis"

_ijms, 2022, doi:10.3390/ijms23052789_

Round 1
Reviewer 1 Report
This study investigates the role of the antimicrobial peptide (AMP) Def2 from Coridius chinensis. The authors use bioinformatics to identify important nucleotides, ORFs, stop codon, cleavage sites in the CcDEF2 gene and use PyMOL modeling to predict the structure, cationic regions and S-S bonds in the post-translated peptide. They used MEGA X to perform a MSA and create a phylogeny of AMPs in insects using plants and mouse AMPs as roots. RT-PCR was used to determine the expression levels of CcDef2 during egg, nymph, and adult stages in various tissues, concluding that 5th instar fat bodies had the highest expression levels. Expression levels were then measured in bacterially challenged vs. control adult insects at various time points and specifically in fat bodies. Expression levels peak at 12hr post-innoculation and return to control levels after 45 hours. The authors transformed CcDef2 into E.coli and were able to express and identify the ~20kDa trans-protein using SDS-Page and western blotting. Petri dishes with gram negative and gram positive bacteria containing a recombinant CcDef2 AMP inclusion body in a center hole, showed that CcDef2 had antimicrobial properties specifically against gram-positive bacteria such as Streptococcus aureus, Bacillus subtilis, and M. luteus. The AMP was not effective in reducing growth by gram-negative bacteria. The authors also did a minimal inhibitory concentration test and found that just 6ug/L of peptide was enough to significantly reduce growth. Overall the paper reads well and is of interest to the reader.
My comments are listed below.
**The most important comment I have is regarding Line 407 in the methods. "Healthy adult samples without any treatment were used as the control group." Were control insects given intraperitoneal injections of PBS buffer without bacteria? If not, then you can not include section 2.4 or figure 7 in the manuscript as it would be impossible to delineate whether the injection or the bacteria caused the increase in expression of CcDef2.
Other comments include:
-Lines 74-76 are a sentence fragment.
-Line 82, replace the word proved with "shown"
-Lines 88-95, Can you draw a schematic of what the RNA transcript looks like including the UTRs, ORFs, introns, and stop codon? This would be good for visualization (in addition to FIgure 2.
Line 110, I am also not sure it is necessary to show the gel of the PCR product in Figure 1.
-LInes 103-104, You state that the peptide has a substrate-binding groove among the alpha helix and beta sheets. What does this bind to? Is there enzymatic function to the protein or is this where it binds to the negatively charged bacterial membrane (is this where the 6 basic amino acids are)? This just needs some clarity.
Figure 4: Put an arrow or indicate where the AT at the N terminus and the R at the C-terminus is on in the figure. The print is small and it is hard to see the letters so having this pointed out would be preferred.
-Figure 5: Can you put boxes around the CcDef genes to make them stand out?
-Figure 6a: You have Male and Female columns in the bar graph. Were these taken from whole insect samples or specific male and female parts? Make this clear in the text.
-Lines 205-207. No change in CcDef2 was detected after feeding bacteria...? This is unclear and need to be reworded.
-Figure 7b. Why did you use an ANOVA and Duncan's test when coming only 2 sample groups? Wouldn't a t-test be more appropriate?
-Lines 213-214, This opening sentence is confusing and should be reworded.
-Line 234, This is the first time MIC is appearing in the text, please write it out fully and define it.
**General comment, why are you using "CK" for Control/blank. Can you just use the word control?
Table 1: You indicate that the data are expressed as the mean. How many samples were used (what was your N)? This is not indicated in the methods section either.
**All captions of figures should indicate what the sample size was (how many biological replicates).
The entire discussion section is disorganized and could use additional editing rewording. Here are some more specific comments:
-Lines 264, 285, 321, You need citations/references here.
-Line 290, Can you include in the text what the most common pathogens of C. chinensis are in a natural environment? Are any of the bacteria you used native pathogens for this insect?
The final paragraph in the discussion (lines 334-337) needs to be reworded. The same can be said for the entire "Conclusions" section on page 15.
-Lines 298/and 301 should be switched. This is unclear and does not flow logically.
-Line 314, AMPs as "evolutionary ancient immune weapons". Is this your quote or someone elses? If the latter, cite the source. If it is your quote, own it as it is a cool name.
-
Author Response
Response to Comments and Suggestions for Authors
Reviewer 1:
This study investigates the role of the antimicrobial peptide (AMP) Def2 from Coridius chinensis. The authors use bioinformatics to identify important nucleotides, ORFs, stop codon, cleavage sites in the CcDEF2 gene and use PyMOL modeling to predict the structure, cationic regions and S-S bonds in the post-translated peptide. They used MEGA X to perform a MSA and create a phylogeny of AMPs in insects using plants and mouse AMPs as roots. RT-PCR was used to determine the expression levels of CcDef2 during egg, nymph, and adult stages in various tissues, concluding that 5th instar fat bodies had the highest expression levels. Expression levels were then measured in bacterially challenged vs. control adult insects at various time points and specifically in fat bodies. Expression levels peak at 12hr post-innoculation and return to control levels after 45 hours. The authors transformed CcDef2 into E. coli and were able to express and identify the ~20kDa trans-protein using SDS-Page and western blotting. Petri dishes with gram negative and gram positive bacteria containing a recombinant CcDef2 AMP inclusion body in a center hole, showed that CcDef2 had antimicrobial properties specifically against gram-positive bacteria such as Streptococcus aureus, Bacillus subtilis, and M. luteus. The AMP was not effective in reducing growth by gram-negative bacteria. The authors also did a minimal inhibitory concentration test and found that just 6ug/L of peptide was enough to significantly reduce growth. Overall the paper reads well and is of interest to the reader.
My comments are listed below.
**The most important comment I have is regarding Line 407 in the methods. "Healthy adult samples without any treatment were used as the control group." Were control insects given intraperitoneal injections of PBS buffer without bacteria? If not, then you can not include section 2.4 or figure 7 in the manuscript as it would be impossible to delineate whether the injection or the bacteria caused the increase in expression of CcDef2.
Response: Thanks for your valuable comment. This is a clerical error when we wrote the manuscript. As for bacterial infection assay, one hundred healthy C. chinensis adults were randomly selected for intraperitoneal injection using 1 μL of the bacterial suspension, and healthy adult samples were infected with PBS as the control group. Details of the modifications can be seen in the revised manuscript in Line 505.
Other comments include:
-Lines 74-76 are a sentence fragment.
Response: We had corrected this mistake in the revised manuscript in Lines 73-85.
-Line 82, replace the word proved with "shown"
Response: We appreciate your valuable comment. We have replaced the word proved with "shown" in Line 82 in the revised manuscript.
-Lines 88-95, Can you draw a schematic of what the RNA transcript looks like including the UTRs, ORFs, introns, and stop codon? This would be good for visualization (in addition to Figure 2).
Response: We modified Figures that include Figure 2a and Figure 2b. Figure 2a also has been changed and Figure 2b shows the mRNA transcript of CcDef2. These changes were shown in the manuscript with revision marks. In addition, this sequence is one of the one CcDef2 cDNA, so it does not contain introns.
(a)
(b)
Figure 2. Nucleotide sequence, mRNA, and deduced amino acid sequence of CcDef2. (a) Nucleotide and amino acid sequences of CcDef2. (b) Schematic diagram of the CcDef2 mRNA. The amino acid sequence of the mature CcDef2 is marked by a single underline. The star codon (ATG) is marked by a box. The polyadenylation signal is indicated with a double underline. Asterisk indicates the stop codon (TAG). Vertical arrows indicate the cleavage sites of signal peptide and precursor peptide. Six cysteines are marked with gray shade. The 5′ untranslated region (UTR) contains the 1st to 38th nucleotides and the 3′ UTR contains the 390th to 859th nucleotides.
Line 110, I am also not sure it is necessary to show the gel of the PCR product in Figure 1.
Response: Figure 1 shows an evidence to clone the CcDef2 cDNA, which can make the manuscript more complete and coherent, so we think it may be appropriate to be put in the main text; or this figure can be put in the Supplementary Materials.
-Lines 103-104, You state that the peptide has a substrate-binding groove among the alpha helix and beta sheets. What does this bind to? Is there enzymatic function to the protein or is this where it binds to the negatively charged bacterial membrane (is this where the 6 basic amino acids are)? This just needs some clarity.
Response: CcDef2 is a short peptide that has no enzymatic activity. Six cysteine residues and the eight other conserved amino acids (Ala74, Thr75, Asp77, Ser80, Ala94, Gly104, Gly105, and Arg115) form a substrate-binding groove among the α-helix and β-pleated sheets of the mature CcDef2, which was the predicted binding position with the negatively charged bacterial membrane.
Figure 4: Put an arrow or indicate where the AT at the N terminus and the R at the C-terminus is on in the figure. The print is small and it is hard to see the letters so having this pointed out would be preferred.
Response: We had marked the AT at the N-terminus and the R at the C-terminus in Figure 4 with pounds and a black box, respectively, in the revised manuscript.
Figure 4. Multiple sequence alignment of four defensins from C. chinensis and typical defensins from other species. Pounds indicate alanine (A) and threonine (T), and the black box marks arginine (R).
-Figure 5: Can you put boxes around the CcDef genes to make them stand out?
Response: For “Figure 5. A cluster dendrogram of 36 defensins from 28 species”, four defensins from C. chinensis are marked with different markers, respectively. A filled triangle marks CcDef2, a triangle indicates CcDef3, a circle marks CcDef, and a filled circle indicates CcDef1.
-Figure 6a: You have Male and Female columns in the bar graph. Were these taken from whole insect samples or specific male and female parts? Make this clear in the text.
Response: The adults come from whole insect samples of different male and female individuals. Please kindly see the change in Line279 in the revised manuscript.
-Lines 205-207. No change in CcDef2 was detected after feeding bacteria...? This is unclear and need to be reworded.
Response: The sentence “However, no change in CcDef2 expression was detected after feeding bacteria” has been deleted in the revised manuscript.
-Figure 7b. Why did you use an ANOVA and Duncan's test when coming only 2 sample groups? Wouldn't a t-test be more appropriate?
Response: For the comparison between the two groups, the results of t-test and Duncan's test analyses are the same and showed in Figure 1. It can be seen from Figure 1 below, the variance of the two groups is homogeneity. The test results show that there is a significant difference between the two groups.
(a)
(b)
Figure 1 Results of t-test using SPSS 26.0
-Lines 213-214, This opening sentence is confusing and should be reworded.
Response: The sentence “the nucleotide sequence of the mature CcDef2 was optimized according to the preference of codon usage in E. coli” was changed into “The nucleotide sequence of the mature CcDef2 was optimized according to the preference of codon usage in E. coli, and the optimized sequence is shown in Figure S1”. Please kindly see the change in Lines 303-304 in the revised manuscript.
-Line 234, This is the first time MIC is appearing in the text, please write it out fully and define it.
Response: The full name (minimum inhibitory concentration) of MIC was added in the sentence in Line 337.
**General comment, why are you using "CK" for Control/blank. Can you just use the word control?
Response: We have changed “CK” into “Control” in Figures 7 and 9 in the revised manuscript.
Table 1: You indicate that the data are expressed as the mean. How many samples were used (what was your N)? This is not indicated in the methods section either.
Response: We have described the number of samples and biological duplicates in Lines 540-541 in the Materials and Methods section of the revised manuscript. In addition, please kindly see the sentence in Lines 536-537.
**All captions of figures should indicate what the sample size was (how many biological replicates).
Response: Please kindly see “Values are the mean ± SD of three replicates” (Line 280, in legend of Figure 6). “Data are represented as the mean ± SD of three replicates” was added to the legend of Figure 7 (Line 299). The “of three replicates” was added to legends of Figure 9 (Line 356) and Figure 10 (Line 359), respectively.
The entire discussion section is disorganized and could use additional editing rewording. Here are some more specific comments:
-Lines 264, 285, 321, You need citations/references here.
Response: We have cited relevant references ([32], [48]) in Lines 381 and 442, respectively. As for the statement “At least two defensin homologues are usually found in many sequenced insect genomes or transcriptomes”, it was written by ourselves based on the results of search against the NR database in NCBI, without any reference.
-Line 290, Can you include in the text what the most common pathogens of C. chinensis are in a natural environment? Are any of the bacteria you used native pathogens for this insect?
Response: At present, research on C. chinensis is still at the initial stage, and it is not clear that what the most common pathogens of this insect in a natural environment; next, we will investigate this issue.
The final paragraph in the discussion (lines 334-337) needs to be reworded. The same can be said for the entire "Conclusions" section on page 15.
Response: The final paragraph in Discussion section has been removed. Please kindly see the relative description in the revised manuscript.
-Lines 298/and 301 should be switched. This is unclear and does not flow logically.
Response: These sentences have been modified. Please kindly see the changes in Lines 414-422 in the revised manuscript.
-Line 314, AMPs as "evolutionary ancient immune weapons". Is this your quote or someone else? If the latter, cite the source. If it is your quote, own it as it is a cool name.
Response: We have cited the relevant reference ([46]) in Line 434 in the revised manuscript.
Reviewer 2 Report
In this study, the authors characterized a defensin CcDef2 from Coridius chinensis both through bioinformatics and functional analyses. Identifying antimicrobial peptides such as cecropin from insects remains an important approach to combat the pandemic of antibiotic resistance.
Overall, the authors did a very comprehensive preliminary analysis of CcDef2 in this study and provided a framework for future studies of antimicrobial peptides in C. chinensis. I do have a few questions as follows.
- It was unclear in the text whether the CcDef2 coding sequence was previously identified. Was this ORF previously annotated as CcDef2 or a hypothetical protein?
- Can the author explain why they chose to focus on CcDef2, instead of other CcDef homologs? Do all CcDef homologs co-exist in chinensis?
- Line 206: can authors elaborate on why feeding bacteria to the insect does not stimulate the CcDef2 expression?
- Is there literature reporting the antimicrobial properties of other CcDef homologs?
- Can the authors elaborate on what future experiments should be done to further study the antimicrobial properties of CcDef2?
Author Response
Response to Comments and Suggestions for Authors
Reviewer 2:
In this study, the authors characterized a defensin CcDef2 from Coridius chinensis both through bioinformatics and functional analyses. Identifying antimicrobial peptides such as cecropin from insects remains an important approach to combat the pandemic of antibiotic resistance.
Overall, the authors did a very comprehensive preliminary analysis of CcDef2 in this study and provided a framework for future studies of antimicrobial peptides in C. chinensis. I do have a few questions as follows.
- It was unclear in the text whether the CcDef2 coding sequence was previously identified. Was this ORF previously annotated as CcDef2 or a hypothetical protein?
Response: Thanks for your valuable comment. This is a polypeptide annotated previously in the transcriptome and genome of C. chinensis. CcDef2 shares the same sequence and structural characteristics as most insect defensins. The motif of CcDef2 is C-×16-C-×3-C-×9-C-×4-C-×1-C and is consistent with the typical one of a defensin.
- Can the author explain why they chose to focus on CcDef2, instead of other CcDef homologs? Do all CcDef homologs co-exist in chinensis?
Response: Previous studies have shown that the antibacterial activity of defensins is related to the number of its own charges (Kluver et al, 2005; Lee et al, 2012). So far, our team found four defensin genes of C. chinensis. In view of the above factors, we chose CcDef2 with 5 positive charges. All CcDef homologs co-exist in C. chinensis.
- Line 206: can authors elaborate on why feeding bacteria to the insect does not stimulate the CcDef2 expression?
Response: Through repeated experiments, we found that feeding bacteria can't stimulate the change of CcDef2 expression. It is speculated that there were not enough bacteria to induce immune response in C. chinensis, because insect defensins are a natural immune polypeptides produced by the fat body and hemolymph during injury and invasion of pathogenic microorganisms (Wu, 2008; Qi, 2020).
- Is there literature reporting the antimicrobial properties of other CcDef homologs?
Response: Here are two relevant articles:
(1) Anomala defensins A and B showed potent activity against Gram-positive bacteria, with slight differences in activity against a few strains of tested bacteria(Yamauchi, 2001).
(2) A. luxuriosa defensin 1 exhibited antibacterial activity against both Gram-positive and Gram-negative bacteria(Ueda, et al. 2005.
Actually, our experimental results show that CcDef1 and CcDef3 have antimicrobial activities. These data have not yet been published.
- Can the authors elaborate on what future experiments should be done to further study the antimicrobial properties of CcDef2?
Response: We are planning to further study the functions of CcDef2 with CRISPR/Cas9 technology and its antibacterial mechanisms using electron microscope and cell apoptosis experiments.
Reviewer 3 Report
The manuscript by Gong et al. entitled „Identification and function analysis of a defensin CcDef2 from Coridius chinensis“ presents essential new data about defensin from C. chinensis it expression and function.
This is an interesting manuscript that presents the findings of a well designed and executed research.
Minor remarks:
line 87. Characteristic analysis of CcDef2. Is the CcDef2 sequence deposited in GenBank? What is the acception number?
line 165. Figure 4. There are not accession numbers for alignmed defensins
line 175. Figure 5. Dendrogram would be more precise if it were constructed for Def2 only
line 292. gene duplicationa ...
Author Response
Response to Comments and Suggestions for Authors
Reviewer 3:
The manuscript by Gong et al. entitled “Identification and function analysis of a defensin CcDef2 from Coridius chinensis“ presents essential new data about defensin from C. chinensis it expression and function.
This is an interesting manuscript that presents the findings of a well designed and executed research.
Minor remarks:
- line 87. Characteristic analysis of CcDef2. Is the CcDef2 sequence deposited in GenBank? What is the acception number?
Response: Thanks for your valuable comment. The CcDef2 sequence deposited in GenBank, whose accession number is MN816377. We have added it in the revised manuscript in Line 88.
- line 165. Figure 4. There are not accession numbers for alignmed defensins.
Response: Accession numbers for alignmed defensins had been listed in the revised manuscript in Lines 217-221.
Figure 4. Multiple sequence alignment of four defensins from C. chinensis and typical defensins from other species. These species include seven insects: Sarcophaga peregrina (SpDef1), P18313; Anopheles gambiae (AgDef), Q17017; Protophormia terraenovae (PtDef), P10891; Tenebrio molitor (TmDef), Q27023; Drosophila virilis (DvDef), AHW49172; Drosophila melanogaster (DmDef), P36192 and Apis mellifera (AmDef1), P17722; a plant Vigna radiata (VrDef), AAR08912 and two mammals Mus musculus (MmDef1), NP_ 034161; and MmDef2, NP_001182563). Asterisks indicate positions of cysteines in insect defensins, solid triangles represent positions of cysteines in mammalian defensins, and solid circles denote positions of cysteines in plant defensins. Pound indicate alanine (A) and threonine (T), and black box marks arginine (R).
- line 175. Figure 5. Dendrogram would be more precise if it were constructed for Def2 only
Response: Phylogenetic tree is a method to describe the correlation between different organisms in bioinformatics. For “Figure 5. A cluster dendrogram of 36 defensins from 28 species”, this picture not only shows that the defensins from Hemiptera and Hymenoptera cluster into a clade, but also shows that the four defensins from C. chinensis form two subtypes.
- line 292. gene duplicationa ...
Response: We are sorry for this clerical error and we have removed the letter “a”. Details of the modifications can be seen in Line 369 in the revised manuscript.
Round 2
Reviewer 1 Report
The authors have addressed all of my concerns and comments.